# Experimental Study on the Internal Pressure Pulsation Characteristics of a Bidirectional Axial Flow Pump Operating in Forward and Reverse Directions

Xiaowen Zhang [1], Fangping Tang [1,\*], Yueting Chen [1], Congbing Huang [2], Yujun Chen [3], Lin Wang [1] and Lijian Shi [1]

1   College of Hydraulic Science and Engineering, Yangzhou University, Yangzhou 225009, China; zhangxw_yzu@163.com (X.Z.); chenyt_yzu@163.com (Y.C.); wanglin.yzu@gmail.com (L.W.); shilijian@yzu.edu.cn (L.S.)
2   Jiangsu Aerospace Hydraulic Equipment Co., Ltd., Yangzhou 225009, China; huangcb_jsht@163.com
3   Jiangsu Qinhuai River Water Conservancy Project Management Office, Nanjing 210001, China; chenyj_qhhglc@163.com
\*   Correspondence: tangfp@yzu.edu.cn; Tel.: +86-139-5105-9552

**Abstract:** A bidirectional axial flow pump can realize bidirectional pumping, which has a wide application prospect in coastal low-head pumping stations and water jet propulsion systems. In this paper, a typical bidirectional axial flow pump is taken as the research object, and the hydraulic model of the bidirectional axial flow pump is manufactured. The hydrodynamic characteristics of the bidirectional axial flow pump are tested on the high-precision hydraulic mechanical test bench, including the positive and negative directions. In the experiment, multiple pressure pulsation monitoring points were arranged in the impeller chamber, and the pressure fluctuations in the pump under a total of 42 flow conditions were measured by a micro pressure pulsation sensor, involving 21 working conditions of forward operation and 21 working conditions of reverse operation. According to the experimental results, the hydrodynamic characteristics, especially the pressure pulsation characteristics in the pump, of the two-way axial flow pump under positive and negative operation are comprehensively compared and analyzed, and the energy characteristics and the propagation law of pressure pulsation of the two-way axial flow pump under positive and negative operation are revealed. The research results provide an important reference for the safe and stable operation of coastal bidirectional axial flow pump stations.

**Keywords:** bidirectional axial flow pump; bidirectional operation; experiment; hydrodynamic characteristics; pressure pulsation; spectrum





## 1. Introduction

In recent years, axial flow pumps have gradually played an increasingly important role in water jet propulsion systems and coastal low-head pumping stations [1–4]. At present, most coastal low-head pumping stations adopt axial flow pumps.

In the application of axial flow pump in coastal pumping stations, researchers and managers found that the pumping stations in coastal areas have low head and large water level variations in upstream and downstream. Many coastal low-head pumping stations need bidirectional operation to meet the needs of both drainage and irrigation. There are three main ways to achieve bidirectional operation of coastal pumping stations. The first is equipped with a two-way pumping impeller for direct reverse pumping. The second is a two-way channel. The bidirectional pumping is realized by the special layout of the bidirectional flow channel [5]. The third one is to disassemble the impeller and install it after turning it 180 degrees. Since it is very difficult to disassemble and reinstall impellers frequently in the actual operation of pumping stations, the third method has not been

applied. Compared with the first method, the second method uses the special arrangement of two-way channel to realize two-way pumping, which is expensive and inconvenient to maintain. So the most reliable and convenient way to realize the bidirectional operation of a coastal pumping station is to equip the axial flow pump with bidirectional pumping for direct reverse pumping.

However, due to the one-way impeller basically equipped in the previous axial flow pump station [6–8], research on the hydrodynamic characteristics of the bidirectional axial flow pump is still in a very preliminary stage. The research is mainly aimed at the energy characteristics of the bidirectional axial flow pump. Ma et al. [9] designed a new type of bidirectional axial flow pump by reducing the camber of the curved airfoil and thickening the airfoil skeleton, which significantly improved the hydraulic efficiency and cavitation performance of the bidirectional axial flow pump. Pei et al. [10] studied the influence of the distance between the impeller and the guide vane on the forward and reverse performance of the bidirectional axial flow pump by using the entropy generation method based on the numerical calculation results. Meng et al. [11] used the two-way fluid-solid coupling method to quantitatively analyze the deformation stress of the impeller of the two-way axial flow pump. It was found that the total deformation continued to rise from the inlet to the outlet of the impeller, whether the impeller was in positive or negative rotation. Ma et al. [12] studied the internal flow field of a bidirectional axial flow pump during reverse operation. It was found that the flow separation at the trailing edge of the blade was the main reason for the significant decline of the performance of the bidirectional axial flow pump during reverse operation. Meng et al. [13] optimized the bidirectional axial flow pump based on the method of double-layer artificial neural network. The forward operation efficiency of the bidirectional axial flow pump obtained by the optimized design was reduced, but the reverse operation efficiency was greatly improved.

Although the energy characteristics of the two-way axial flow pump have been improved to some extent, the hydrodynamic performance of the axial flow pump with two-way airfoil is poor, and the inlet flow pattern of the pump is disturbed by the guide vane during the reverse operation. This means the two-way pump in operation will produce more serious vibration and noise [14], and if equipped with a two-way axial flow pump the pumping station system security and stability is hugely at risk. In recent years, more and more researchers have pointed out that the pressure pulsation induced by unstable flow in the pump is the most critical factor affecting the safety and stability of the pumping station system [15–18]. Therefore, it is urgent to study the hydrodynamic characteristics of the bidirectional pump, especially the internal pressure pulsation characteristics.

Considering the high cost and long test cycle of the pressure pulsation experiment [19–21], it is convenient to study the pressure pulsation characteristics of the pump by using numerical simulation [22–25]. However, some existing studies on the two-way pump [26] show that the numerical simulation method is not a particularly reliable means to study the hydrodynamic characteristics of the two-way pump because the vortex, reflux, and other unstable flow phenomena in the two-way pump are obvious under reverse operation. In this paper, the hydrodynamic characteristics of the bidirectional axial flow pump are studied by the experimental method. Firstly, the hydraulic model of a typical two-way axial flow pump was manufactured, and its hydrodynamic characteristics were tested on a high-precision hydraulic mechanical test bench, including positive and negative directions. In the experiment, multiple pressure pulsation monitoring points were arranged in the impeller chamber, and the pressure fluctuations in the pump under a total of 42 flow conditions were measured by using a micro pressure pulsation sensor, involving 21 working conditions of forward operation and 21 working conditions of reverse operation. Based on the experimental results, the hydrodynamic characteristics of the bidirectional axial flow pump under positive and negative operation, especially the pressure pulsation characteristics in the pump, are comprehensively analyzed and compared, and the energy characteristics and the propagation law of pressure pulsation of the bidirectional axial flow pump under positive and negative operation are revealed. The

research results provide an important reference for the safe and stable operation of coastal bidirectional axial flow pump stations.

## 2. Experimental Model and System

### 2.1. Research Object

This experiment takes a typical bidirectional axial flow pump as the research object. The impeller diameter of the bidirectional axial flow pump hydraulic model is 300 mm. The speed is 1450 r/min. The blade placement angle was 0°. The design lift of forward operation is 3.32 m, and that of reverse operation is 4.03 m. The tip clearance is 0.15 mm. The number of impeller blades is four. The impeller blade is formed by numerical control machining with thick copper plate, which avoids the defects such as sand holes and pores caused by casting. The hub body was formed by numerical control machining with copper bars. The number of guide leaves is five. The vanes of guide vanes are processed by mould and welded. Figure 1 shows the experimental figure of the hydraulic model of the bidirectional axial flow pump. Figure 2 shows the rotation direction of the impeller of the bidirectional axial flow pump.

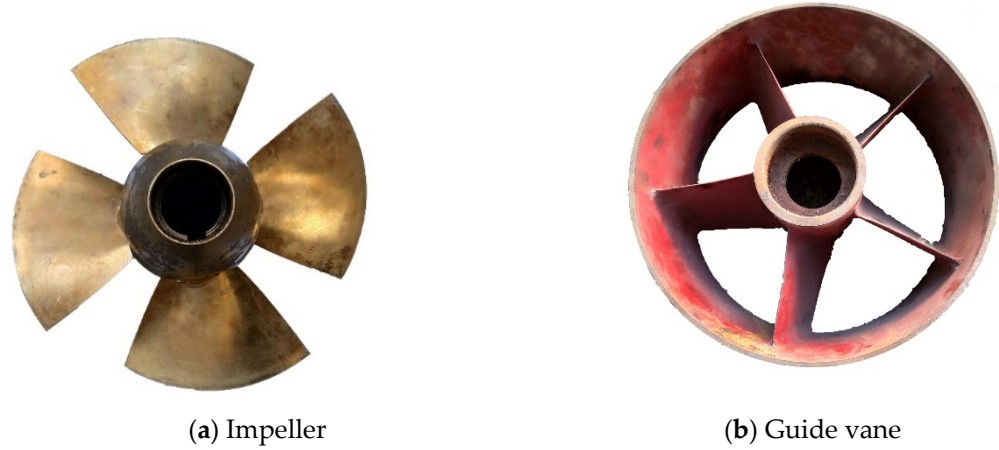

(**a**) Impeller      (**b**) Guide vane

**Figure 1.** Hydraulic model of bidirectional axial flow pump.

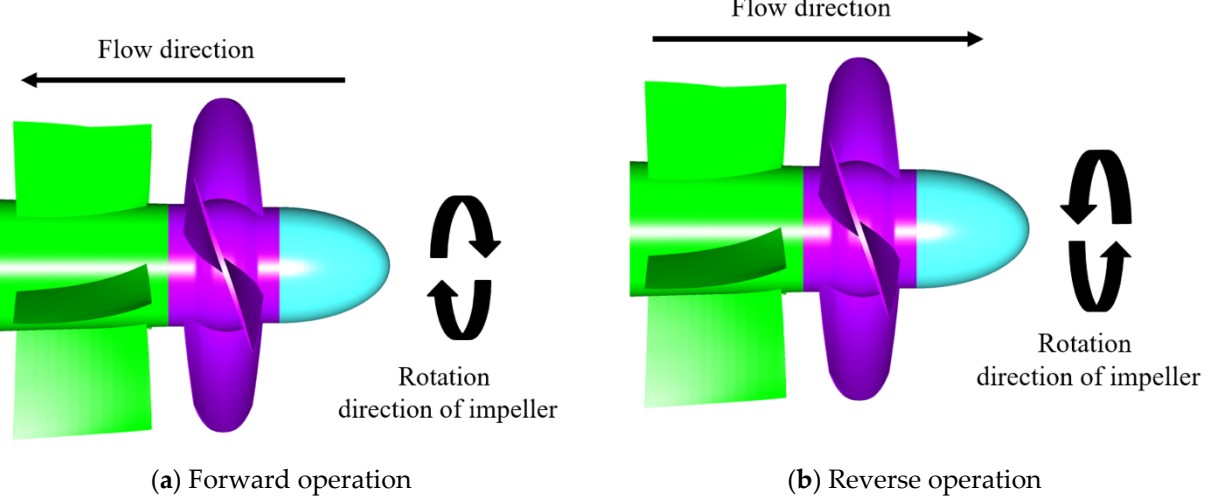

(**a**) Forward operation      (**b**) Reverse operation

**Figure 2.** Rotation direction of impeller of bidirectional axial flow pump.

### 2.2. Experimental Model

The experimental test of the standard pump section was carried out on the hydraulic model of the bidirectional pump completed by processing. The standard pump section included straight pipe section, impeller, guide vane, and 60° elbow. Figure 3 shows the

schematic diagram of the standard pump section in the experiment. Figure 4 shows the schematic diagram of the experimental pump section after installation and splicing.

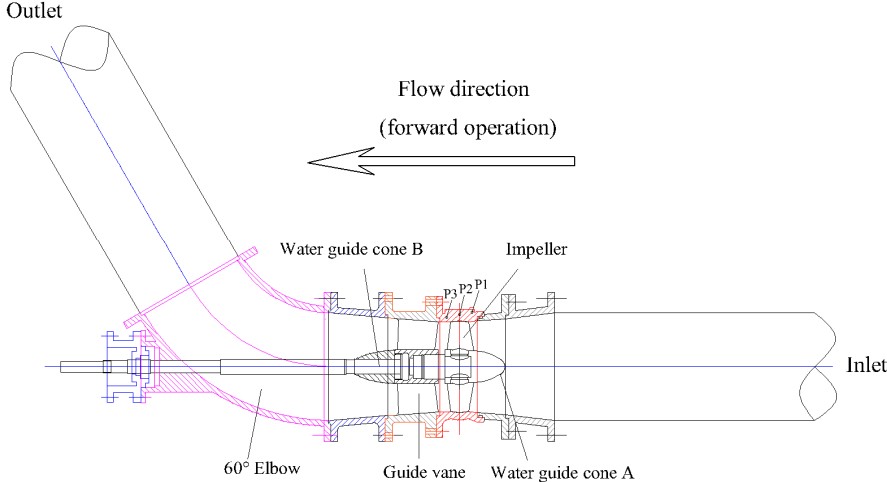

(**a**) Forward operation

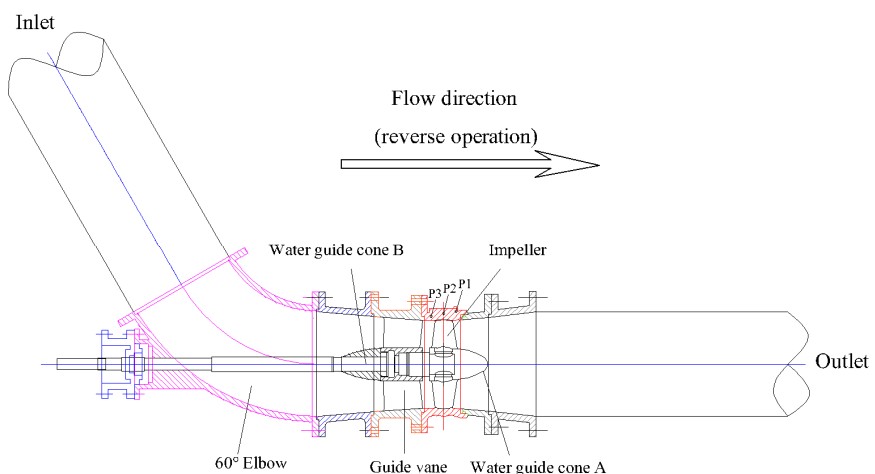

(**b**) Reverse operation

**Figure 3.** Schematic diagram of standard pump section in experiment.

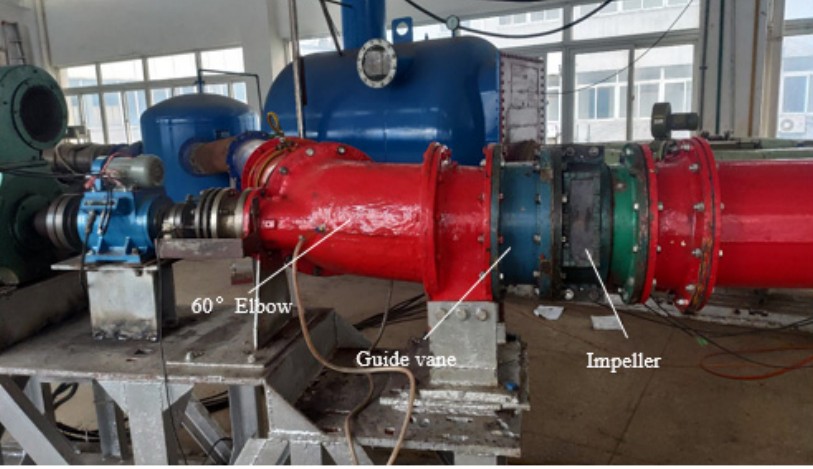

**Figure 4.** Schematic diagram of the experimental pump section after installation and splicing.

*2.3. Experimental Test System*

In this paper, a high-precision hydraulic mechanical test bench is built. The standard pump section equipped with the bidirectional pump hydraulic model was tested on the test bench, including energy characteristics experiment and pressure pulsation characteristics experiment. During the test, strict reference was made to 'water pump model and device model acceptance test procedures' (SL 140-2006) requirements [27–29]. The high-precision hydraulic mechanical test bench is a large vertical sealing system. The experimental system includes forward and reverse operation, control gate valve, auxiliary pump unit, working condition adjustment gate valve, intake tank, pressure outlet tank, stable pressure rectifier cylinder, etc. The schematic diagram of the experimental system is shown in Figure 5. The test accuracy of the test system is 0.39%. The system uncertainty of pump performance test is the square and root of each single system uncertainty. The calculation formula is as follows [30]:

$$(E_\eta)_s = \pm\sqrt{E_Q^2 + E_H^2 + E_M^2 + E_n^2} = \pm 0.274\% \tag{1}$$

where $E_Q$ is the system uncertainty of flow measurement and the calibration result is $\pm 0.2\%$. $E_H$ is the uncertainty of the static head measurement system, the calibration results of the full range of $\pm 0.10\%$. $E_M$ is the system uncertainty of torque measurement, the uncertainty of torque speed sensor is $\pm 0.15\%$. $E_n$ is the system uncertainty of speed measurement. When the sampling period is 2 s and the speed is not less than 1000 r/min, the uncertainty is $\pm 0.05\%$.

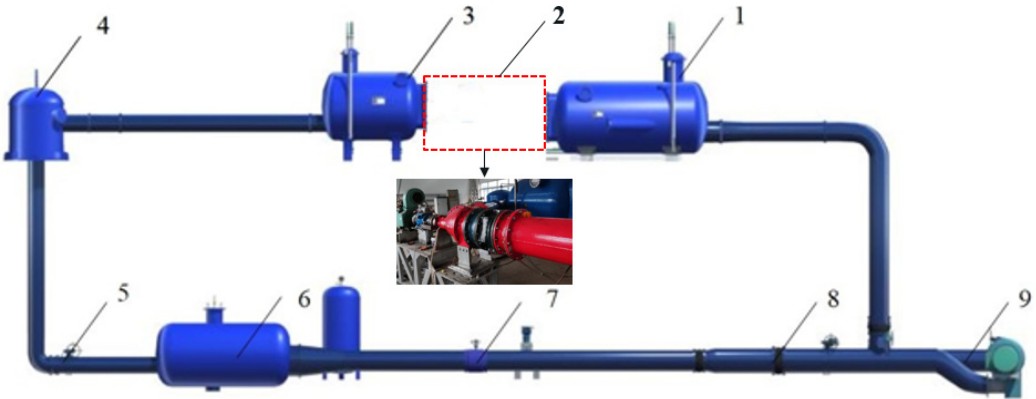

1. Intake tank. 2. Tested pump unit and drive motor. 3. Pressure outlet tank. 4. Bifurcation tank. 5. Condition regulating gate valve. 6. Voltage regulating rectifier. 7. Electromagnetic flowmeter. 8. System forward and reverse operation control gate valve. 9. Auxiliary pump unit.

**Figure 5.** Physical schematics of the test bench.

The main measuring instruments of the test system include differential pressure transmitter, electromagnetic flowmeter, torque meter, speed torque sensor, absolute pressure transmitter. and so on. Table 1 shows the specific parameters of the main measuring instruments for the energy characteristic test. Table 2 shows the specific parameters of pressure sensor in pressure pulsation test. The specific parameters of the pressure sensor in the pressure pulsation test are shown in Table 2. The sampling frequency of the pressure pulsation sensor is 124 times that of the impeller rotation frequency (RF), which can meet the requirements of the pressure pulsation acquisition in the axial flow pump system [31]. The pressure pulsation experiment uses synchronous channel record, monitoring points p1, p2, and p3 pressure pulsation signal acquisition is synchronous. Figure 6 shows the schematic layout of pressure pulsation measuring points. Three pressure fluctuation monitoring points are located near the impeller inlet, the impeller middle, and near the impeller outlet.

**Table 1.** Main measuring instruments for energy characteristic experiment.

| Measuring Items | Instrument Name | Instrument Types | Instrument Range | Calibration Accuracy |
|---|---|---|---|---|
| Head | Difference pressure transmitter | EJA 110A | 0~200 kPa | ±0.1% |
| Flow | Electromagnetic flowmeter | E-mag type | DN400 mm | ±0.20% |
| Torque | Torsiograph | JW-3 | 200 Nm | ±0.15% |
| Rotation speed | The speed and torque sensor | JC | 0~10,000 r/min | ±0.15% |

**Table 2.** The specific parameters of the pressure sensor used for the pressure field measurement in the pump.

| Item | Parameter |
|---|---|
| Model | HM90A |
| Precision | 0.1% |
| Range | 0~200 kPa |
| Output signal | 0~5 V |
| Sampling frequency | 3 kHz |

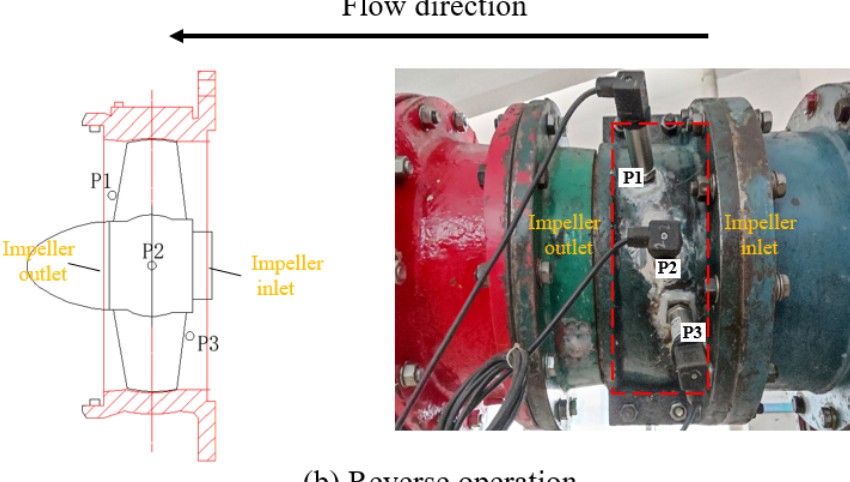

(a) Forward operation

(b) Reverse operation

**Figure 6.** Schematic layout of pressure pulsation measuring points.

## 3. Experimental Results of Energy Characteristics

The bidirectional axial flow pump was experimentally tested on a high-precision hydraulic mechanical test bench, including forward operation and reverse operation. The test speeds of forward and reverse operation condition were both 1450 r/min. The flow rate of the pump defining the highest efficiency point of the forward operation condition is $Q_{bep1}$, and the flow rate of the pump defining the highest efficiency point of the reverse operation condition is $Q_{bep2}$. The experimental energy characteristic curve of bidirectional axial flow pump is shown in Figure 7. The characteristic parameters of the highest efficiency point under forward operation condition and reverse operation condition are shown in Table 3.

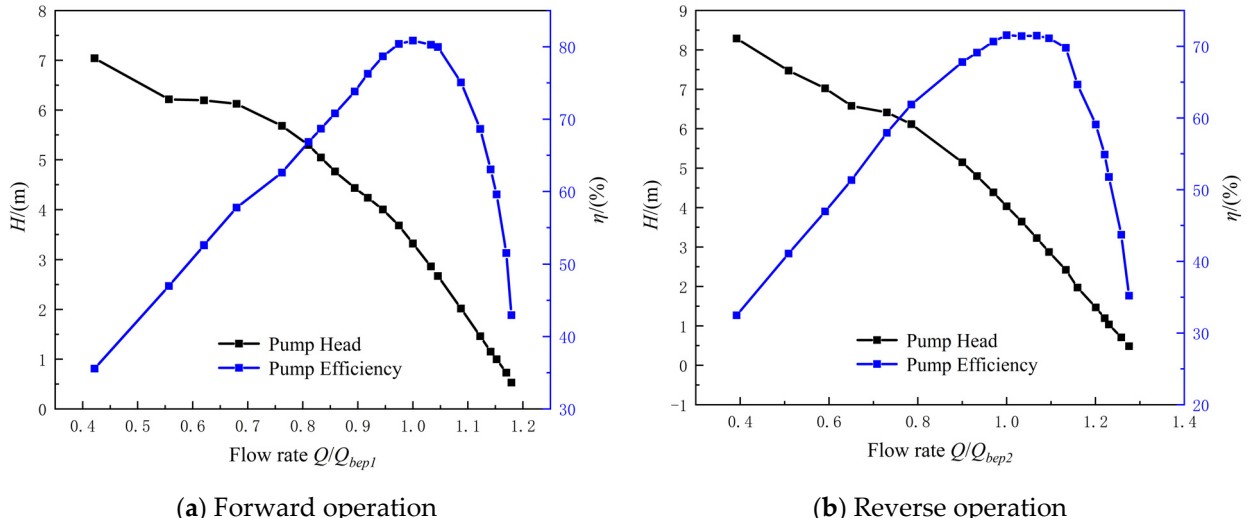

(**a**) Forward operation                              (**b**) Reverse operation

**Figure 7.** Experimental energy characteristic curve of bidirectional axial flow pump.

**Table 3.** Characteristic parameters of highest efficiency points for forward and reverse operation.

| Parameters at the Highest Efficiency Point $Q_{bep}$ | Forward Operation Condition | Reverse Operation Condition |
|---|---|---|
| Flow $Q_d$ | 376.53 | 308.05 |
| Head $H$ (m) | 3.32 | 4.03 |
| Shaft power $N$ (kW) | 15.13 | 16.98 |
| Efficiency $\eta$ (%) | 80.85 | 71.55 |

The variation trend of efficiency and head with flow rate of bidirectional axial flow pump in reverse operation is similar to that in forward operation. The efficiency increases first and then decreases with the increase of flow rate, and the head decreases gradually with the increase of flow rate. Under the forward operation condition, the flow rate at the highest efficiency point of the bidirectional axial flow pump is 376.53 L/s, the head is 3.32 m, the axial power is 15.13 kW, and the efficiency is 80.85%. Under the reverse operation condition, the flow rate at the highest efficiency point of the bidirectional axial flow pump is 308.05 L/s, the head is 4.03 m, the axial power is 16.98 kW, and the efficiency is 71.55%.

In order to more intuitively compare the energy characteristics of forward and reverse operating conditions, the conversion coefficient K is introduced. By comparing the flow, head, efficiency at the highest efficiency point, and the flow range in the high efficiency zone (limited by the drop of 5% at the highest efficiency point) of the forward and reverse operating conditions, the ratio of the flow, head, efficiency, and the flow range in the high-efficiency zone of the reverse operating condition is obtained, namely, the conversion coefficient K. The comparison of energy characteristics between forward and reverse operating conditions based on conversion coefficient is shown in Figure 8. It can be seen from Figure 8 that under the reverse operation condition, the flow and efficiency corresponding to the optimal point of the bidirectional axial flow pump are smaller than

those under the forward operation condition. The flow rate is about 81.81% of the forward operation, which is reduced by 18.19%. The efficiency is about 88.50% of the forward operation, which is reduced by 11.50%. Under the reverse operation condition, the range of the high efficiency zone and the corresponding head of the optimum of the bidirectional axial flow pump are greater than those of the forward operation condition. The head is about 121.39% of the forward operation condition, increased by 21.39%. The range of high efficiency zone is about 131.25% of the forward operation condition, increased by 31.25%. From the perspective of energy characteristics, the efficiency of bidirectional axial flow pump in reverse operation is lower than that in forward operation, but the range of high-efficiency area is large. In general, the bidirectional axial flow pump can ensure good hydraulic performance in both forward and reverse operation, and can meet the needs of bidirectional pumping.

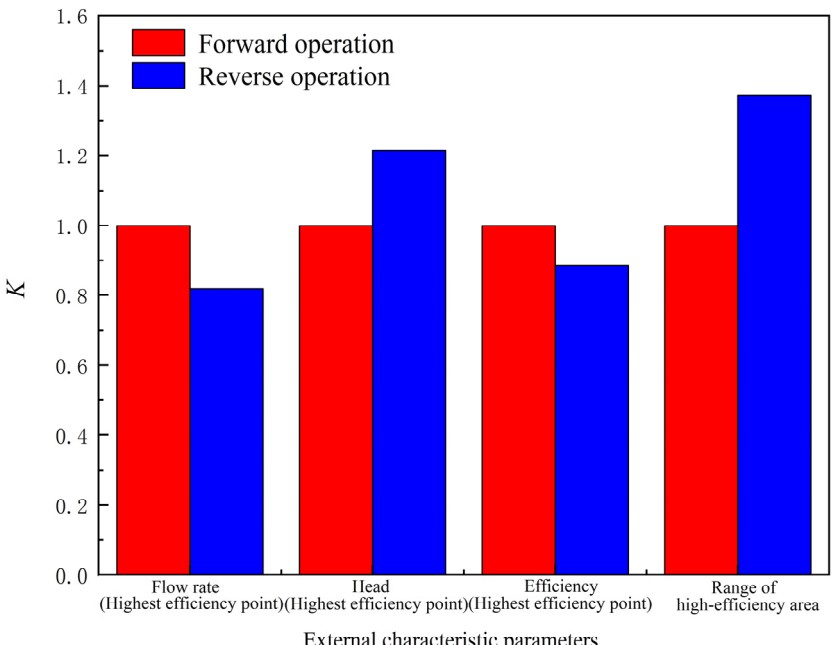

**Figure 8.** Comparison of energy characteristics between forward and reverse operation conditions based on conversion coefficient.

## 4. The Experimental Results and Analysis of Pressure Pulsation Characteristics

### 4.1. Time Domain Analysis of Pressure Fluctuation

In the experiment, the micro pressure pulsation sensor was used to measure the internal pressure fluctuations of the bidirectional axial flow pump under a total of 42 flow conditions, involving 21 forward operating conditions and 21 reverse operating conditions. In order to compare and analyze the pressure fluctuation characteristics of bidirectional axial flow pump under forward and reverse operation conditions, the time domain analysis of pressure fluctuation in pump under different flow conditions is carried out. In the time domain analysis, the pressure fluctuation coefficient is introduced, and the pressure fluctuation in four impeller rotation cycles (one impeller rotation cycle is about 0.041 s) is dimensionlessly processed to eliminate the static pressure interference. The pressure pulsation coefficient formula is as follows [31]:

$$C_p = \frac{p - \overline{p}}{0.5\rho u_2^2} \qquad (2)$$

where $p$ is the transient pressure value. $\overline{p}$ is the average pressure value. $u_2$ is the impeller outlet circumferential velocity.

The experimental data of pressure pulsation at 16 working conditions (including eight working conditions of forward operation and eight working conditions of reverse operation) from $0.4Q_{bep}$ to $1.2Q_{bep}$ are selected for processing. Figure 9 shows the peak-to-peak value of the pressure pulsation coefficient at the monitoring points in the bidirectional axial flow pump under different flow conditions. It can be seen from Figure 9 that no matter whether the bidirectional axial flow pump is in the forward or reverse operation, the maximum value of the peak-to-peak value of the pressure pulsation coefficient in the pump always appears at the inlet of the impeller, and the minimum value of the peak-to-peak value of the pressure pulsation coefficient in the pump appears at the outlet of the impeller in most cases. By comparing Figure 9a,b, it can be found that the variation trend of the peak value of the pressure fluctuation at the inlet of the impeller with the flow rate is, basically, the same under the forward and reverse operation conditions. The variation trend of the peak-to-peak value of the pressure fluctuation at the middle and outlet of the impeller with the flow rate is similar under the small flow condition, and there are some differences near the large flow condition. Compared with the unidirectional axial flow pump, the peak value of the maximum pressure fluctuation at the key position of the bidirectional axial flow pump is slightly offset. According to the experimental results of the pressure pulsation of the unidirectional axial flow pump in the reference [31], the maximum peak-to-peak value of pressure pulsation coefficient of conventional unidirectional axial flow pump appears in the vicinity of $0.6Q_{bep}$–$0.7Q_{bep}$. In this experiment, the maximum peak-to-peak value of pressure pulsation coefficient of bidirectional axial flow pump appears in the vicinity of $0.8Q_{bep}$–$0.9Q_{bep}$ at the key positions under the forward and reverse operation. The maximum peak-to-peak value of the pressure pulsation coefficient at the inlet of the impeller appears near $0.90Q_{bep}$ under the forward and reverse operation conditions, and the maximum peak-to-peak value of the pressure pulsation coefficient at the middle and outlet of the impeller appears near $0.80Q_{bep}$.

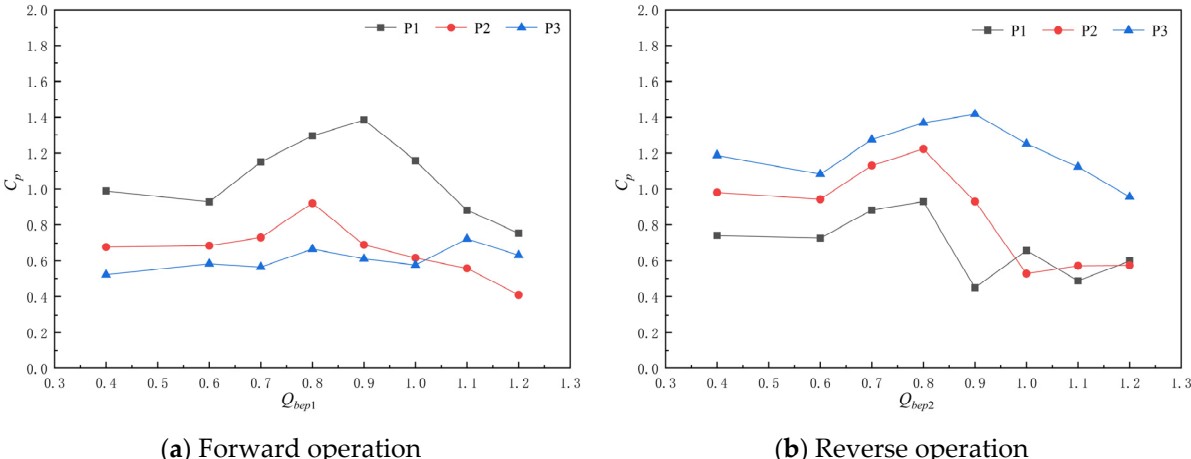

(**a**) Forward operation   (**b**) Reverse operation

**Figure 9.** Peak-to-peak value of the pressure pulsation coefficient in the monitoring points of bidirectional axial flow pump under different flow conditions.

Taking $0.70Q_{bep}$, $1.00Q_{bep}$, and $1.20Q_{bep}$ as typical operating points, the time domain analysis of pressure fluctuation at six operating points under forward and reverse operation conditions was carried out. The time domain diagram of pressure fluctuation coefficient at each monitoring point under forward operation condition is given in Figure 10. Figure 11 shows the pressure fluctuation coefficient time domain diagram of each monitoring point under reverse operation condition. It can be seen from Figures 10 and 11 that under the two-way operation condition, the regularity of the waveform of the pressure pulsation in the two-way axial flow pump is good under the optimal working condition of $1.00Q_{bep}$ and the large flow condition of $1.20Q_{bep}$. There are obviously four main wave peaks and four main wave troughs at each monitoring point in the pump during one impeller rotation

cycle. Under 0.70 $Q_{bep}$ small flow conditions, the two-way axial flow pump in the forward and reverse operation of the impeller inlet reflux and tip leakage vortex and other unstable flow, leading to the phenomenon of vortex induced harmonics in the pump. The regularity of pressure fluctuation waveform of each monitoring point in the pump is obviously poor, and more sub-peak amplitude appears in an impeller rotation cycle.

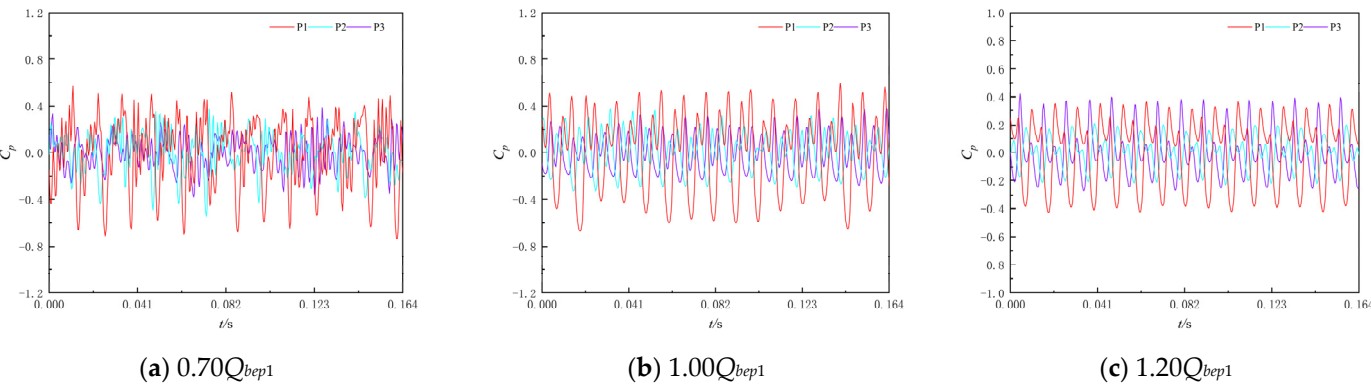

**(a)** 0.70$Q_{bep1}$　　　　　　　　　　**(b)** 1.00$Q_{bep1}$　　　　　　　　　　**(c)** 1.20$Q_{bep1}$

**Figure 10.** Time domain diagram of pressure fluctuation coefficient of each monitoring point under forward operation condition.

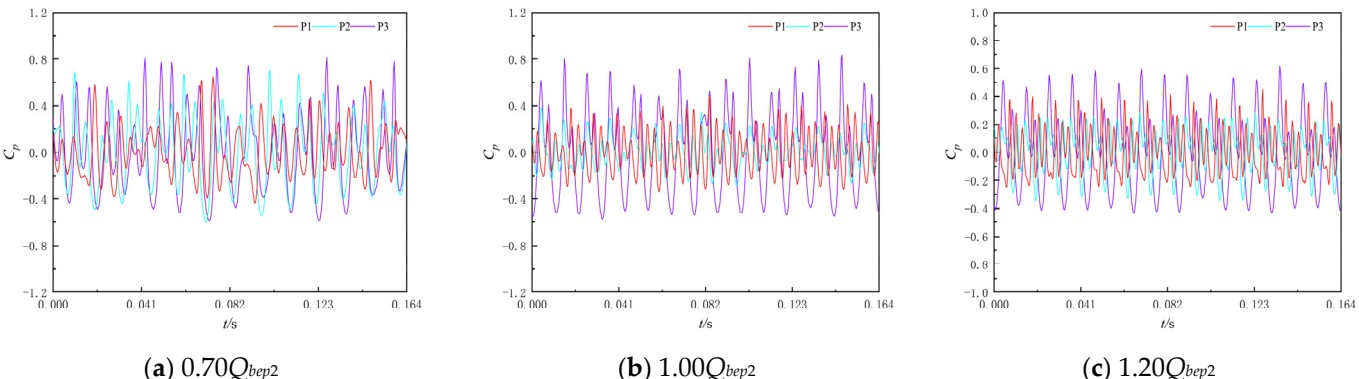

**(a)** 0.70$Q_{bep2}$　　　　　　　　　　**(b)** 1.00$Q_{bep2}$　　　　　　　　　　**(c)** 1.20$Q_{bep2}$

**Figure 11.** Time domain diagram of pressure fluctuation coefficient of each monitoring point under reverse operation condition.

*4.2. Frequency Domain Analysis of Pressure Fluctuation at the Highest Efficiency Point*

In order to analyze the local characteristics of the pressure pulsation signal and accurately identify the frequency component of the pressure pulsation, the pressure pulsation data are processed by fast Fourier transform (FFT). In order to ensure that the Fourier transform has enough high resolution, 3000 sampling points are selected for Fourier transform. The frequency resolution of this experiment is 1 Hz, and the definition of frequency resolution is as follows [31]:

$$\Delta f = \frac{f_s}{M} \tag{3}$$

where $f_s$ is the sampling frequency of the pressure sensor, 3000 Hz. $M$ is the number of sample points selected number, 3000 in total.

The data in the four rotation cycles of the impeller are taken for fast Fourier transform analysis. The X axis of the transverse axis of the spectrum diagram is the rotation frequency multiple, the Y axis is the different monitoring points, and the Z axis of the longitudinal axis is the pressure pulsation amplitude. The frequency conversion multiple formula is as follows [31]:

$$N_F = \frac{f}{f_n} = \frac{60F}{n} \tag{4}$$

where $F$ is the frequency after Fourier transform. $n$ is impeller speed.

Figure 12 shows the pressure pulsation distribution of the bidirectional axial flow pump at the optimal point of $1.00Q_{bep}$. The following conclusions can be drawn from Figure 12. Firstly, the pressure pulsation of the bidirectional axial flow pump at the optimum point of $1.00Q_{bep}$ is dominated by the blade frequency (BPF = 96.67 Hz) or the high-order harmonics of the blade frequency. In forward operation, the main frequency of pressure pulsation at impeller inlet monitoring point P1 is blade frequency, the main frequency of pressure pulsation at impeller middle monitoring point P2 and impeller outlet monitoring point P3 is twice blade frequency (2BPF = 193.33 Hz). In reverse operation, the main frequency of pressure pulsation at the impeller inlet monitoring point P3 and the impeller central monitoring point P2 is blade frequency, and the main frequency of pressure pulsation at the impeller outlet monitoring point P1 is three times blade frequency (3BPF = 290.00 Hz). It can be found that the results of the main frequency components of the pressure pulsation of the bidirectional axial flow pump obtained in this experiment are different from the experimental conclusions obtained in the reference that the pressure pulsation of the key monitoring points of the unidirectional axial flow pump is controlled by the blade frequency [31]. The research shows that when the bidirectional axial flow pump operates under the internal design conditions, in addition to the frequency components caused by the impeller rotation cycle, there are other high-frequency components, such as noise caused by fluid excitation and static and dynamic interference in the pump [32–34].

Secondly, under the optimal condition of $1.00Q_{bep}$, the composition of the pressure pulsation signal in the pump is relatively simple, and the pulsation of each monitoring point in the low-frequency region and high-frequency region is small. The pulsating components in the low-frequency region are mainly concentrated in the axial frequency (SF = 24.17 Hz). The pulsation in high-frequency region is mainly concentrated in the high order harmonics of the blade frequency, but it is almost not observed in the high-frequency region above four times the blade frequency. Thirdly, the pressure pulsation frequency components of the impeller inlet are very close in the forward and reverse operation conditions, and the pressure pulsation frequency components in the middle of the impeller change greatly. The main frequency of the monitoring points in the middle of the impeller changes from two times the blade frequency in the forward operation to the blade frequency in the reverse operation, and the secondary main frequency changes from the blade frequency to two times the blade frequency in the reverse operation.

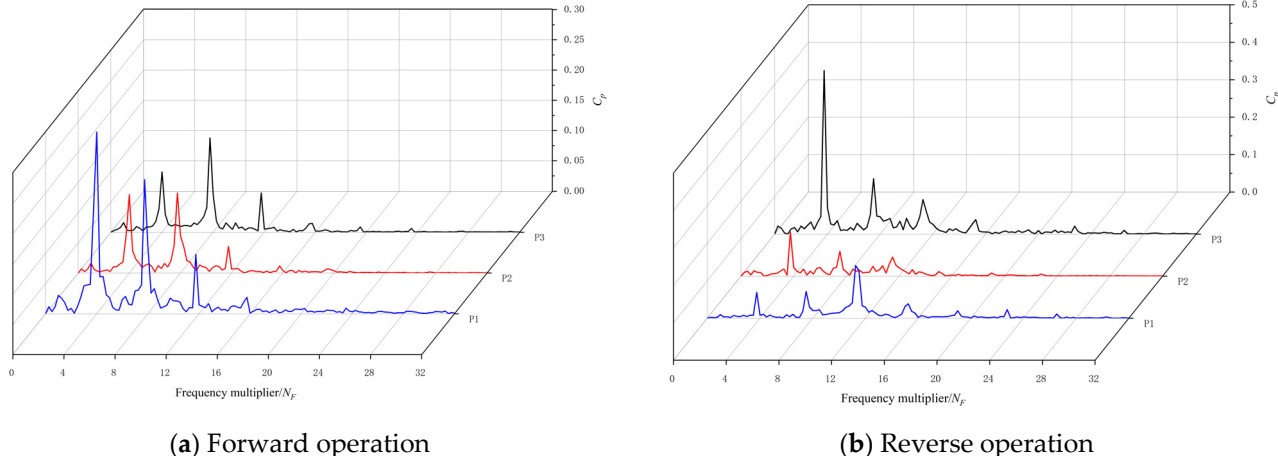

**(a)** Forward operation　　　　　　　　　　　　　　　　　　**(b)** Reverse operation

**Figure 12.** Pressure pulsation distribution of bidirectional axial flow pump in optimal condition of $1.00Q_{bep}$.

Figure 13 shows the $C_p$ amplitude of the main frequency of the pressure pulsation of the bidirectional axial flow pump under the optimal condition of $1.00Q_{bep}$. The following conclusions can be drawn from Figure 13. Firstly, no matter whether the bidirectional axial flow pump is in the forward or reverse operation, the $C_p$ amplitude of the main

frequency of the pressure pulsation in the pump decreases first and then increases from the impeller inlet to the impeller outlet. The $C_p$ amplitudes from the impeller inlet to the outlet are 0.30, 0.13, and 0.16, respectively, under the forward operation condition. Under the reverse operation condition, the $C_p$ amplitudes from the impeller inlet to outlet are 0.44, 0.12 and 0.14, respectively. The maximum value of $C_p$ amplitude in both forward and reverse operation occurs at the inlet of the impeller, which is due to the suction of the impeller in the large range of low-pressure area formed by the suction surface of the blade. There is also a local impact on the flow and the inlet edge of the impeller. The velocity gradient and pressure gradient of the fluid particles are also large, and the final rotating pressure gradient shows a large pressure pulsation. Secondly, no matter whether the bidirectional axial flow pump is in the forward or reverse operation, the maximum $C_p$ amplitude of the main frequency of the pressure pulsation in the pump appears at the inlet of the impeller, and the minimum $C_p$ amplitude appears at the middle of the impeller. The $C_p$ amplitude of the main frequency of the pressure pulsation at the inlet of the impeller is 2.31 times that at the outlet of the impeller under the positive rotation operation condition, and it reaches 3.67 times under the reverse operation condition. Thirdly, compared with the forward operation condition, the $C_p$ amplitude of the main frequency of the pressure pulsation at the inlet of the impeller under the reverse operation condition increases by 46.67%, the central part of the impeller decreases by 7.69%, and the outlet of the impeller decreases by 12.50%.

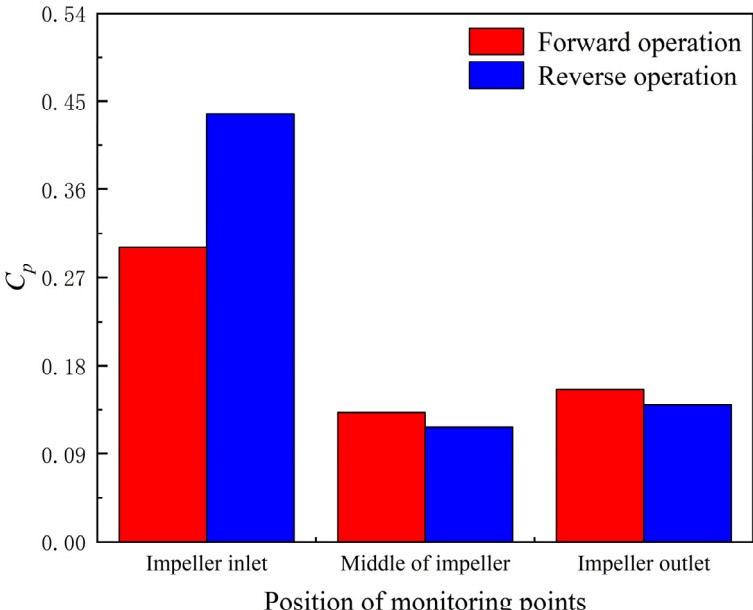

**Figure 13.** $C_p$ amplitude of main frequency of pressure pulsation of bidirectional axial flow pump under optimal condition of $1.00Q_{bep}$.

### 4.3. Frequency Domain Analysis of Pressure Pulsation under Small Flow Conditions

Figure 14 shows the pressure pulsation distribution of the bidirectional axial flow pump at a small flow rate of $0.70Q_{bep}$. The following conclusions can be drawn from Figure 14. Firstly, the pressure fluctuation of the bidirectional axial flow pump at a small flow rate of $0.70Q_{bep}$ is dominated by the blade frequency (BPF = 96.67 Hz) or the higher order harmonics of the blade frequency. In the positive operation, the main frequency of pressure pulsation at each monitoring point in the pump is the blade frequency. In reverse operation, the main frequency of pressure fluctuation at impeller inlet monitoring point P3 and impeller middle monitoring point P2 is blade frequency, and the main frequency of pressure fluctuation at impeller outlet monitoring point P1 is twice blade frequency (2BPF = 193.33 Hz). Secondly, the low-frequency signals induced by impact and reflux are very rich in the bidirectional axial flow pump at small flow rate of $0.70Q_{bep}$. The pulsations

of different frequency components can be observed in the low-frequency region, whether in the forward or reverse operation. This phenomenon is related to the rotational stall in the pump under small flow conditions, indicating that low-frequency local vortices appear at the inlet and outlet of the impeller.Thirdly, compared with the reverse operation, the frequency band of the pressure fluctuation spectrum is significantly wider in the forward operation, and the pulsation amount in the high-frequency region is significantly increased. A certain amount of pulsation can still be observed at 4BPF~7BPF. The reason may be that under the condition of small flow in forward operation, there is a relatively unstable vortex area in the pump, which causes the water flow in the pump to cause different degrees of impact on the walls of the blades and the flow channel, forming pressure waves of different frequencies.

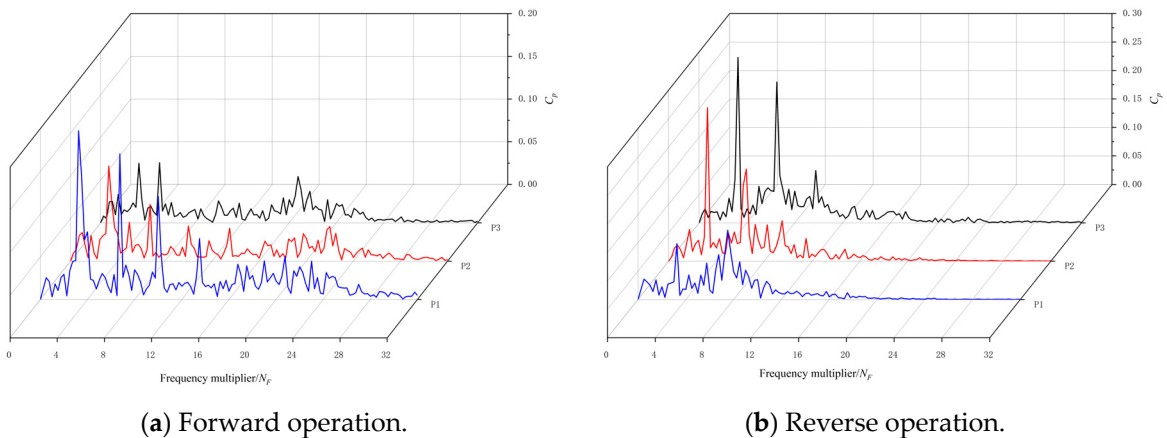

(**a**) Forward operation.　　　　　　(**b**) Reverse operation.

**Figure 14.** Pressure pulsation distribution of bidirectional axial flow pump in small flow rate of $0.70Q_{bep}$.

　　Figure 15 shows the $C_p$ amplitude of the main frequency of the pressure pulsation of the bidirectional axial flow pump at a small flow rate of $0.70Q_{bep}$. The following conclusions can be drawn from Figure 15. Firstly, no matter whether the bidirectional axial flow pump is in the forward or reverse operation, the $C_p$ amplitude of the main frequency of the pressure pulsation in the pump decreases gradually from the impeller inlet to the impeller outlet. The $C_p$ amplitudes from the impeller inlet to outlet are 0.20, 0.11, and 0.07, respectively, under the forward operation condition. Under the reverse operation condition, the $C_p$ amplitudes from the impeller inlet to outlet are 0.29, 0.27 and 0.12, respectively. Secondly, no matter whether the bidirectional axial flow pump is in the forward or reverse operation, the maximum value of $C_p$ amplitude of the main frequency of pressure pulsation in the pump appears at the impeller inlet, and the minimum value appears at the impeller outlet. Under the forward operation condition, the $C_p$ amplitude of the main frequency of the pressure pulsation at the impeller inlet is 2.86 times that at the impeller outlet, and the reverse operation condition reached 2.42 times. Third, compared with the forward running condition, the $C_p$ amplitude of the main frequency of pressure pulsation at each monitoring point is larger under the reverse running condition. The $C_p$ amplitude of the main frequency of the pressure fluctuation at the inlet of the impeller increased by 45.00%, the central part of the impeller increased by 145.45%, and the outlet of the impeller increased by 71.43%. It indicates that the energy conversion of water flow in the pump is relatively intense under the condition of reverse operation with small flow rate.

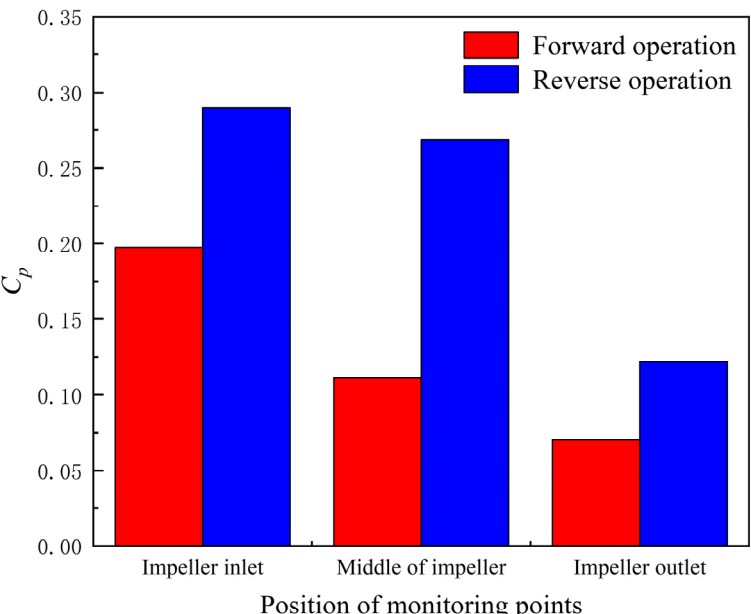

**Figure 15.** $C_p$ amplitude of main frequency of pressure pulsation of bidirectional axial flow pump at small flow rate of $0.70Q_{bep}$.

### 4.4. Frequency Domain Analysis of Pressure Pulsation under Large Flow Conditions

Figure 16 shows the pressure pulsation distribution of the bidirectional axial flow pump at a large flow rate of $1.20Q_{bep}$. The following conclusions can be drawn from Figure 16. Firstly, the pressure pulsation of the bidirectional axial flow pump at large flow rate of $1.20Q_{bep}$ is dominated by the high-order harmonic of the blade frequency (BPF = 96.67 Hz) or the blade frequency. In forward operation, the main frequency of pressure fluctuation at impeller inlet monitoring point P1 and impeller middle monitoring point P2 is blade frequency, and the main frequency of pressure fluctuation at impeller outlet monitoring point P3 is twice blade frequency (2BPF = 193.33 Hz). In reverse operation, the main frequency of pressure fluctuation at impeller inlet monitoring point P3 and impeller middle monitoring point P2 is blade frequency, and the main frequency of pressure fluctuation at impeller outlet monitoring point P1 is four times blade frequency (4BPF = 386.68 Hz). Secondly, the pressure fluctuation spectrum of the bidirectional axial flow pump under the condition of large flow rate of $1.20Q_{bep}$ has few frequency components except the frequency doubling of the blade frequency, and the pressure fluctuation signal component tends to be simple. This shows that the flow pattern of the bidirectional axial flow pump is relatively good under the condition of large flow rate, and the unstable flow phenomenon is relatively weak. Thirdly, compared with the forward operation, a certain amount of pulsation can be observed in the low-frequency and high-frequency regions of the pressure pulsation spectrum at the middle and outlet of the impeller under the reverse operation. The occurrence of pulsation in low-frequency region is related to the fact that there is no guide vane recycling ring at the outlet of the impeller under reverse operation. The flow of high velocity near the middle and outlet of the impeller is unstable, resulting in a certain amount of pulsation in the low-frequency region of the pressure pulsation spectrum at the middle and outlet of the impeller. The occurrence of the pulsation in the high-frequency region may be due to the fact that under the condition of large flow rate, the axial flow pump is more likely to induce the early generation of cavitation in the reverse direction, and the occurrence of cavitation pulsation leads to large pulsation in the reverse pressure pulsation spectrum [32–34].

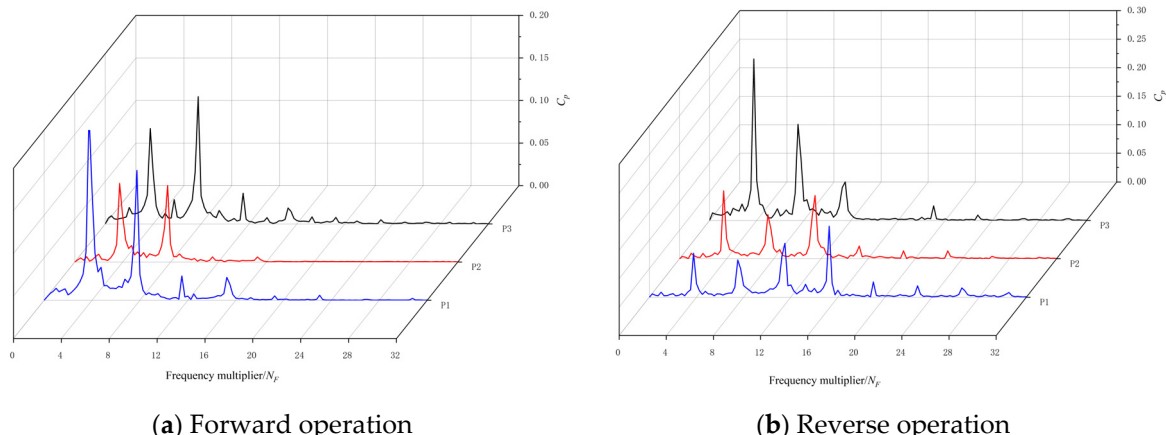

(**a**) Forward operation　　　　　　　　　　(**b**) Reverse operation

**Figure 16.** Pressure pulsation distribution of bidirectional axial flow pump in large flow rate of 1.20$Q_{bep}$.

Figure 17 shows the $C_p$ amplitude of the main frequency of the pressure pulsation of the bidirectional axial flow pump at a large flow rate of 1.20$Q_{bep}$. The following conclusions can be drawn from Figure 17. Firstly, no matter whether the two-way axial flow pump is running in the forward or reverse direction, the change rule of the Cp amplitude of the main frequency of the pressure pulsation in the pump is the same, which first decreases and then increases from the impeller inlet to the impeller outlet. The $C_p$ amplitudes from the impeller inlet to the outlet are 0.22, 0.09, and 0.15, respectively, under the forward operation condition. Under the reverse operation condition, the $C_p$ amplitudes from the impeller inlet to outlet are 0.28, 0.12, and 0.13, respectively. Secondly, no matter whether the bidirectional axial flow pump is in the forward or reverse operation, the maximum value of $C_p$ amplitude of the main frequency of pressure pulsation in the pump appears at the inlet of the impeller, and the minimum value appears at the middle of the impeller. The $C_p$ amplitude of the main frequency of the pressure pulsation at the inlet of the impeller under the forward operation condition is 1.47 times that at the outlet of the impeller, and the reverse operation condition reaches 2.15 times. Thirdly, compared with the forward operation condition, the $C_p$ amplitude of the main frequency of the pressure pulsation at the inlet of the impeller under the reverse operation condition increases by 27.27%, the central part of the impeller increases by 33.33%, and the outlet of the impeller decreases by 13.33%.

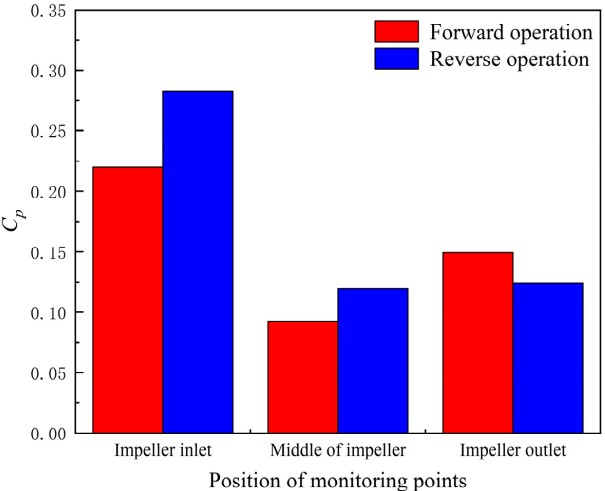

**Figure 17.** $C_p$ amplitude of main frequency of pressure pulsation of bidirectional axial flow pump at large flow rate of 1.20$Q_{bep}$.

### 4.5. Frequency Domain Analysis of Pressure Pulsation under Different Flow Conditions

Figure 18 shows the pressure pulsation distribution of monitoring points P1 and P3 in the pump under different operating conditions. The following conclusions can be drawn from Figure 18. Firstly, no matter whether the two-way axial flow pump is running in the forward or reverse direction, the main frequency of pressure pulsation at impeller inlet is always dominated by the blade frequency (BPF = 96.67 Hz). It shows that the rotation of the impeller is the main cause of the pressure pulsation at the inlet of the bidirectional pump impeller. Secondly, when running in the forward direction, the main frequency of the pressure pulsation at the impeller outlet under different flow conditions is always dominated by twice the impeller frequency (2BPF = 193.33 Hz). During reverse operation, with the increase of flow rate, the main frequency of pressure pulsation at the impeller outlet has a tendency to gradually move to the high-frequency region, from two times the impeller frequency under the $0.70Q_{bep2}$ flow condition, to the $1.00Q_{bep2}$ flow condition three times the blade frequency (3BPF = 290.00 Hz) at the time, and then transfer to four times the blade frequency (4BPF = 386.68 Hz) when the flow rate is $1.20Q_{bep2}$. Thirdly, when running in forward rotation, as the flow rate increases, the significant pulsation in the high-frequency region of the impeller inlet and outlet gradually disappears. In reverse operation, the pressure pulsation at the impeller outlet is sensitive to the change of flow rate. With the increase of flow rate, the pulsation in the high-frequency region increases. It shows that with the increase of flow rate, the unstable flow phenomenon in the pump gradually disappears in the forward operation, and the unstable flow such as cavitation in the pump tends to be severe under the condition of large flow rate in the reverse operation [35–37].

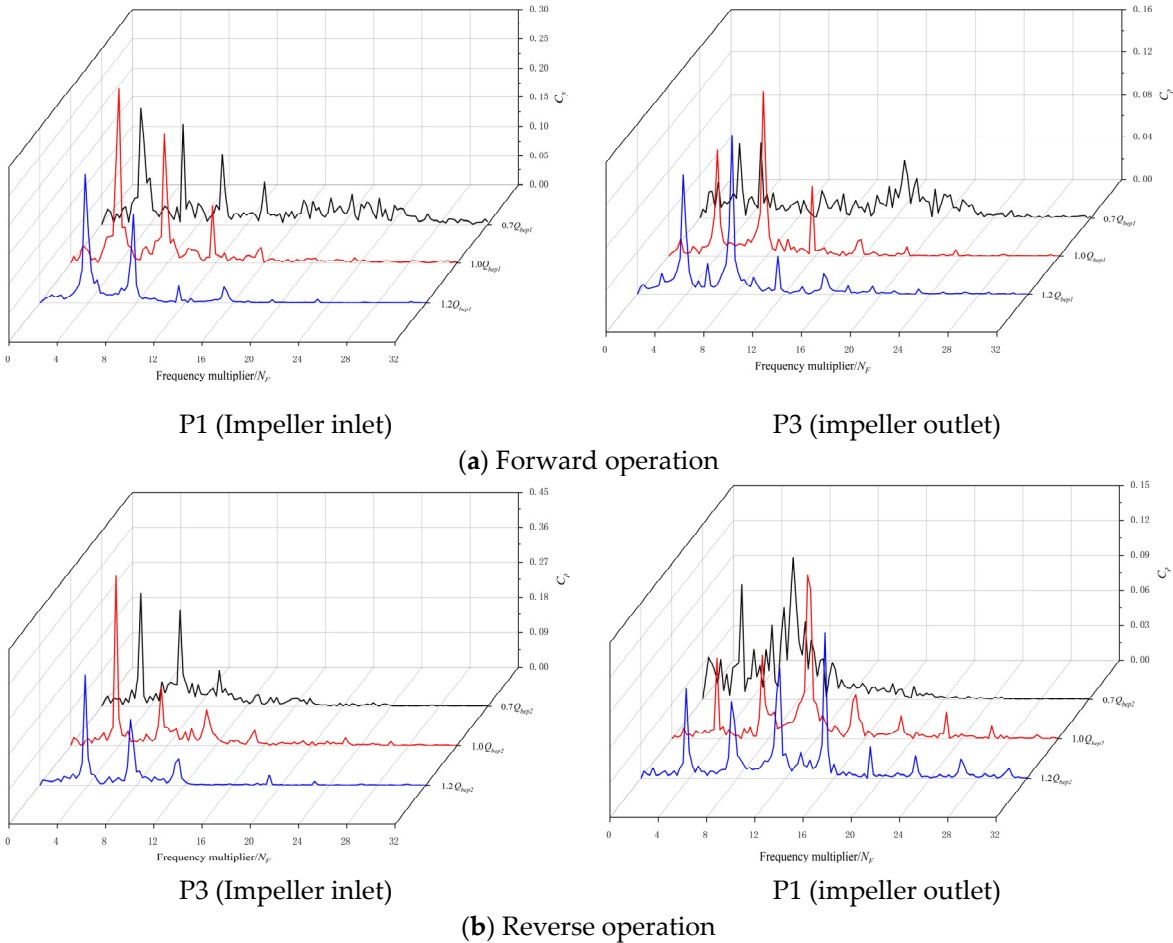

**Figure 18.** Pressure pulsation distribution of monitoring points P1 and P3 in pump under different operating conditions.

Figure 19 shows the main frequency $C_p$ amplitude of the impeller inlet and outlet of the bidirectional axial flow pump under different operating conditions. The following conclusions can be drawn from Figure 19. Firstly, the $C_p$ amplitude of the main frequency of the impeller inlet in the reverse operation condition is greater than that in the forward operation condition under different flow conditions, and the percentage of the difference between the two changes with the flow rate is relatively small, which is always maintained at about 30–45%. Secondly, the $C_p$ amplitude of the main frequency at the outlet of the impeller is very sensitive to the change of the flow rate at the forward operation. As the flow rate increases from $0.70Q_{bep1}$ to $1.20Q_{bep1}$, the $C_p$ amplitude of the main frequency at the outlet of the impeller increases from 0.29 times of 1.15 times of the reverse operation condition. Thirdly, the main frequency $C_p$ amplitude of the impeller inlet and outlet in the forward and reverse operation conditions of the bidirectional axial flow pump has the same trend with the flow rate, which increases first and then decreases with the increase of the flow rate. In forward operation, compared with the optimal $1.00Q_{bep1}$ condition, the $C_p$ amplitude of the impeller inlet main frequency under the condition of small flow $0.70Q_{bep1}$ decreases by 34.14% and the outlet decreases by 54.68%. Under the condition of large flow $1.20Q_{bep1}$, the $C_p$ amplitude of the impeller inlet main frequency decreased by 26.70% and the outlet decreased by 3.78%. In reverse operation, compared with the optimal $1.00Q_{bep2}$ condition, the $C_p$ amplitude of the impeller inlet main frequency under the condition of small flow $0.70Q_{bep2}$ decreases by 33.63% and the outlet decreases by 13.09%. Under the condition of large flow $1.20Q_{bep2}$, the $C_p$ amplitude of the impeller inlet main frequency decreased by 35.36%, and the outlet decreased by 11.32%.

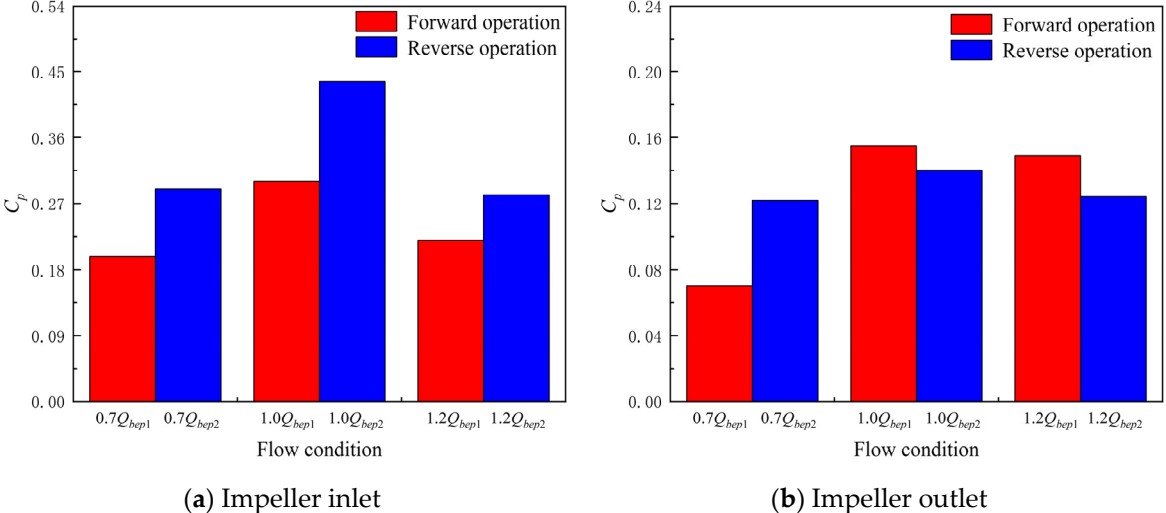

(**a**) Impeller inlet                                      (**b**) Impeller outlet

**Figure 19.** $C_p$ amplitude of main frequency at impeller inlet and impeller outlet of bidirectional axial flow pump under different operating conditions.

## 5. Conclusions

In order to explore the hydrodynamic characteristics of bidirectional axial flow pump, this paper carried out experiments on a bidirectional axial flow pump on a high-precision hydraulic mechanical test bench, including positive and negative directions. In the experiment, a micro pressure pulsation sensor was used to measure the pressure fluctuation in the pump under a total of 42 flow conditions, involving 21 forward operation conditions and 21 reverse operation conditions. Based on the experimental results, the hydrodynamic characteristics of the two-way axial flow pump under forward and reverse operation, especially the pressure pulsation characteristics in the pump, are comprehensively analyzed and compared. The main conclusions are as follows:

(1) Compared with the forward operation condition, the flow rate and efficiency corresponding to the optimal point under the reverse operation condition of bidirectional axial

flow pump are smaller than those under the forward operation condition, and the flow rate is reduced by 18.19%. The efficiency decreased by 11.50%. However, the range of head and high efficiency zone is larger than that of forward operation. The head increased by 21.39% and the scope of the efficient zone increased by 31.25%. From the point of view of energy characteristics, the hydraulic performance of bidirectional axial flow pump is relatively balanced in forward and reverse operation, which can meet the needs of bidirectional pumping.

(2) Under the optimal condition of $1.00Q_{bep}$, the composition of the pressure pulsation signal in the pump is simple, and the pressure pulsation signal in the pump is mainly controlled by the blade frequency (BPF = 96.67 Hz), two times the blade frequency (2BPF = 193.33 Hz) or three times the blade frequency (3BPF = 290.00 Hz). This is different from the previous conclusion that the pressure pulsation of the key monitoring points of the unidirectional axial flow pump is controlled by the blade frequency. Compared with the optimal $1.00Q_{bep1}$ working condition of forward operation, the Cp amplitude of the main frequency of the pressure pulsation at the inlet of the impeller under the optimal $1.00Q_{bep2}$ working condition of reverse operation increases by 46.67%, the central part of the impeller decreases by 7.69%, and the outlet of the impeller decreases by 12.50%.

(3) In forward operation, with the increase of flow rate, the obvious pulsation in the high-frequency region of the impeller inlet and outlet gradually disappears. In reverse operation, the pressure pulsation at the impeller outlet is sensitive to the change of flow rate. With the increase of flow rate, a certain degree of pulsation gradually appears in the high-frequency region. It shows that with the increase of flow rate, the unstable flow phenomenon in the pump gradually disappears in the forward operation, and the energy conversion and flow pattern change at the impeller outlet tend to be intense in the reverse operation.

(4) The main frequency $C_p$ amplitude of the impeller inlet and outlet in the forward and reverse operation conditions of the bidirectional axial flow pump has the same trend with the flow rate, which increases first and then decreases with the increase of the flow rate. In forward operation, compared with the optimal $1.00Q_{bep1}$ condition, the $C_p$ amplitude of the impeller inlet main frequency under the condition of small flow $0.70Q_{bep1}$ decreases by 34.14% and the outlet decreases by 54.68%. Under the condition of large flow $1.20Q_{bep1}$, the $C_p$ amplitude of the impeller inlet main frequency decreased by 26.70% and the outlet decreased by 3.78%. In reverse operation, compared with the optimal $1.00Q_{bep2}$ condition, the $C_p$ amplitude of the impeller inlet main frequency under the condition of small flow $0.70Q_{bep2}$ decreases by 33.63% and the outlet decreases by 13.09%. Under the condition of large flow $1.20Q_{bep2}$, the $C_p$ amplitude of the impeller inlet main frequency decreased by 35.36%, and the outlet decreased by 11.32%.

The current work is mainly to reveal the hydrodynamic characteristics of the bidirectional axial flow pump through experimental methods, especially the pressure pulsation characteristics in the pump. By comparing the forward operation condition with the reverse operation condition, the safety and stability of the bidirectional axial flow pump system in the bidirectional utilization are evaluated. The research results can provide important reference for safe and stable operation of bidirectional axial flow pump station system in bidirectional operation. However, how to eliminate or improve the pressure pulsation in the bidirectional axial flow pump has not been well solved. In the further study, more physical analysis of bidirectional axial flow pump should be carried out based on CFD method to reveal the damage mechanism of pressure pulsation on bidirectional axial flow pump station under bidirectional operation conditions.

**Author Contributions:** Formal analysis and writing original draft preparation, X.Z.; validation and writing review and editing the paper, F.T. and C.H.; data curation, Y.C. (Yujun Chen) and L.S.; paper translation, Y.C. (Yueting Chen) and L.W. All authors have read and agreed to the published version of the manuscript.

**Funding:** This research work was supported by the National Natural Science Foundation of China (funder: National Natural Science Foundation of China. funding number: 51376155), the Natural Science Foundation of Jiangsu Province (funder: Jiangsu Provincial Department of Science and

Technology. funding number: BK20190914), the China Postdoctoral Science Foundation (funder: China Postdoctoral Science Foundation. funding number: 2019M661946), and the University Science Research Project of Jiangsu Province (funder: People's Government of Jiangsu Province. funding number: 19KJB570002), the Water Conservancy Science and Technology Projects of Jiangsu Province (funder: Department of Water Resources of Jiangsu Province. funding number: 2021012).

**Data Availability Statement:** Not applicable.

**Acknowledgments:** A project funded by the Priority Academic Program Development (PAPD) of Jiangsu Higher Education Institutions Support for construction and assembly of the facility was also provided by the Hydrodynamic Engineering Laboratory of Jiangsu Province.

**Conflicts of Interest:** The authors declare no conflict of interest.

## Nomenclature

| | |
|---|---|
| $C_P$ | Pressure pulsation coefficient |
| f | Frequency (Hz) |
| F | Frequency after Fourier transform (Hz) |
| g | Local acceleration of gravity (m/s$^2$) |
| $H_{exp}$ | Experimental head (m) |
| L | Hydraulic loss coefficient |
| K | Conversion coefficient |
| M | The impeller torque (N·m) |
| n | Rated speed (r/min) |
| $N_F$ | Multiple of rotational frequency |
| P | Transient pressure (Pa) |
| $\bar{p}$ | Average pressure (Pa) |
| Q | Flow rate of the model pump system (m$^3$/s) |
| $Q_{bep}$ | Optimal condition |
| t | Time (s) |
| $u_2$ | Circumferential velocity of impeller outlet (m/s) |
| ρ | The density of flow (kg/m$^3$) |
| ω | The angular velocity of the impeller (rad/s) |
| η | Efficiency (%) |
| $η_{exp}$ | Experimental Efficiency (%) |
| **Abbreviations** | |
| CFD | Computational fluid dynamics |
| SF | Shaft frequency |
| SST | Shear–stress transport |
| BPF | Impeller rotation frequency |

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
