# Peer review of "Experimental Study on the Internal Pressure Pulsation Characteristics of a Bidirectional Axial Flow Pump Operating in Forward and Reverse Directions"

_machines, doi:10.3390/machines10030167_

Round 1

Reviewer 1 Report

  1. Please give more specific parameters of the bidirectional axial flow pump, including specific speed, impeller hub diameter, impeller axial height and other parameters.

  1. Please supplement the rotation direction of the impeller under forward and reverse operating conditions.

  1. Please add citations to all equations in the text.

  1. Does the sampling frequency of the pressure pulsation sensor used in the experiment meet the requirements? How to ensure that the details of the changes in the internal flow field of the bidirectional axial flow pump can be captured?

  1. What is the consideration for the selection of the pressure pulsation monitoring point layout in the experiment? Is there any relevant basis?

  1. For the convenience of comparison, it is recommended to use the same scale for the forward and reverse corresponding working conditions in Figure 8 and Figure 9.

  1. In Figure 7, it is found that the maximum pressure pulsation peak-to-peak value at the impeller inlet under forward and reverse operation conditions appears around 0.90Qbep, and the maximum pressure pulsation peak-to-peak value at the impeller middle and impeller outlet appears around 0.80Qbep. Is this consistent with the propagation law of pressure pulsations of conventional unidirectional axial flow pumps?

  1. Is the pressure pulsation characteristic of the two-way axial flow pump different from that of the conventional one-way axial flow pump? A discussion is suggested in the Conclusions section as well as elsewhere in the manuscript, comparing findings with other authors who have published work on conventional unidirectional axial flow pumps. In addition, are the findings of this paper really applicable to the study of other bidirectional axial flow pumps?

  1. The authors should discuss the limitations of the current study and possible improvements for future research work.

In general, the research methods and work of this paper are relatively substantial and innovative, but lack of conciseness, the author is recommended to further improve and modify. I recommend this paper to be accepted after revision.

Author Response

Revision explanation

First of all, on behalf of my co-authors, I would like to thank the editors and reviewers for their valuable and constructive comments on our manuscript ' Experimental study on the internal pressure pulsation characteristics of a bidirectional axial flow pump operating in forward and reverse directions '.

We carefully study the reviewers ' comments and modify the manuscript one by one according to each of the comments made by reviewers.

For the comments of reviewers, we modify and supplement the content of the article, and annotate it with red font. Thank the editors and reviewers once again for their valuable comments.

Responses to the comments of reviewer 1

  1. We have added more specific two-way axial flow pump parameters, including specific speed, impeller hub diameter, impeller axial height and so on. 9. (The contents modified and added in this article have been marked in red fonts to highlight the modification traces)

  1. We have supplemented the rotating direction of the impeller under forward and reverse operating conditions. (The contents modified and added in this article have been marked in red fonts to highlight the modification traces)

  1. We have added citations to all equations in the main text. (The contents modified and added in this article have been marked in red fonts to highlight the modification traces)

  1. The frequency of the pressure pulsation sensor used in this experiment is 3000 Hz. During the pressure pulsation, the pressure fluctuation in a impeller rotation cycle is collected 124 times, and the sampling frequency is 124 times of the impeller rotation frequency (RF). According to the relevant references cited in this paper, the sampling frequency of this experiment can meet the requirements of the pressure pulsation acquisition in the axial flow pump.
  2. In the experiment, the pressure pulsation monitoring points are arranged at the inlet, middle and outlet of the impeller, which can capture the propagation law of the pressure pulsation in the bidirectional axial flow pump along the axial direction, analyze the evolution of the internal flow field of the bidirectional axial flow pump with the flow, and reveal the pressure pulsation amplitude at the key position of the bidirectional axial flow pump. At the same time, the pressure pulsation monitoring points in this paper are also in line with the arrangement of conventional pressure pulsation monitoring points in axial flow pump pressure pulsation experiments.
  3. We have used the same scale for the forward and reverse operating conditions in Figs. 8, 9 and 10. (The contents modified and added in this article have been marked in red fonts to highlight the modification traces)

  1. Compared with the unidirectional axial flow pump, the peak value of the maximum pressure fluctuation at the key position of the bidirectional axial flow pump is slightly offset. According to the experimental results of the pressure pulsation of the unidirectional axial flow pump in the references, the maximum pressure pulsation peak value of the conventional unidirectional axial flow pump appears in the range of 0.6Qbep ~ 0.7Qbep. In this experiment, the maximum pressure pulsation peak value of the key position of the bidirectional axial flow pump in the forward and reverse operation occurs in the range of 0.8Qbep~ 0.9Qbep.
  2. The pressure pulsation characteristics of bidirectional axial flow pump are different from those of unidirectional axial flow pump. We have conducted supplementary discussions in the conclusion part and other parts of the manuscript, and compared the research results with those of other authors who have published the research results of traditional unidirectional axial flow pump.(The contents modified and added in this article have been marked in red fonts to highlight the modification traces).In addition, the local characteristics of pressure pulsation of bidirectional axial flow pumps with different airfoils may be different, but the propagation law and evolution characteristics of pressure pulsation obtained in this paper can provide reference for other bidirectional axial flow pumps.
  1. At the end of the paper, we have discussed the limitations of current research and possible improvements in future research. (The contents modified and added in this article have been marked in red fonts to highlight the modification traces)

Reviewer 2 Report

Equation (2) defines the "pressure fluctuation coefficient" established in terms of publications [27-29]. After examining these references, the introduction of this criterion is not clear. At first glance, this is a cavitation number, where the value of the saturated vapor pressure in the numerator is replaced by the time-averaged value of the instantaneous pressure. In the denominator, the fluid flow velocity is replaced by the circumferential speed of the impeller blades. What is the significance of this criterion when the experiments were performed at constant speed (1450 rpm)?

The results of the experiment in Figure 8 do not show "the peak-to-peak value of the pressure pulsation  in the pump" but "the peak-to-peak value of the pressure pulsation cofficient". (also Figures 9 and 10).

In the frequency domain, the signal was, according to the authors, processed "through Fourier transform". The Fourier transform is applicable to signals that are continuous in time, then the right-hand side of Equation (2) is valid. However, by measuring the time course by pressure, vectors of discrete values were created, representing the amplitude of the pressures at the corresponding time point, to which the Discrete-time STFT is applied.

The results of the frequency domain analysis are shown in one graph. This representation would be correct if the recordings of pressures p1, p2, p3 were realized synchronously. Information on which DAQ system was used is missing: with simultaneous channel recording, or channel recording via multiplexer. The x-axis in the graphic outputs is given incorrectly.

Sampling frequency information and the resulting information on the frequency domain transformation spectrum and frequency analysis resolution are also missing.

Commenting on the results of the analysis in the frequency domain for the highest flow, the fact that the magnitude of the "pressure fluctuation coefficient" increases towards higher harmonic components is neglected. Couldn't cavitation have already taken place in this area?

Author Response

Revision explanation

First of all, on behalf of my co-authors, I would like to thank the editors and reviewers for their valuable and constructive comments on our manuscript ' Experimental study on the internal pressure pulsation characteristics of a bidirectional axial flow pump operating in forward and reverse directions '.

We carefully study the reviewers ' comments and modify the manuscript one by one according to each of the comments made by reviewers.

For the comments of reviewers, we modify and supplement the content of the article, and annotate it with red font. Thank the editors and reviewers once again for their valuable comments.

Responses to the comments of reviewer 2

  1. The introduction of pressure fluctuation coefficient is to compare the pressure fluctuation of different flow conditions at the same level. It is the most recognized and widely used normalized processing method for the analysis of internal pressure fluctuation of hydraulic machinery, especially pump [1-6]. Many articles published in Journal of Fluids Engineering and other top journals of fluid machinery have used this method for the analysis of pressure fluctuation characteristics [1-3].

  1. Thanks to the reviewers for their comments, we have corrected the names of figures 8, 9 and 10 and the relevant statements in the manuscript. (The contents modified and added in this article have been marked in red fonts to highlight the modification traces)

  1. Thanks to the reviewers for their guidance and recommendations, we have replaced and supplemented the formula in the manuscript. (The contents modified and added in this article have been marked in red fonts to highlight the modification traces)

  1. This pressure pulsation experiment adopts synchronous channel recording, and the pressure pulsation signal acquisition of monitoring points P1, P2 and P3 is realized synchronously. According to the recommendations of reviewers, we have made a supplementary explanation in this paper. (The contents modified and added in this article have been marked in red fonts to highlight the modification traces)

  1. We have added the sampling frequency information, frequency resolution and the number of selected sample points to the text according to the comments of the reviewers. The frequency domain transform spectrum is the frequency domain diagram of the pressure fluctuation in the manuscript (Figure 12, 14, 16, 18). (The contents modified and added in this article have been marked in red fonts to highlight the modification traces)

  1. Thanks to the reviewer's correction, after combining the reviewer ' s opinions and consulting relevant references [7], we found that the high frequency pulsation during reverse operation may indeed come from the influence of cavitation pulsation under large flow conditions. So we corrected previous comments on the results of the frequency domain analysis of the highest traffic. (The contents modified and added in this article have been marked in red fonts to highlight the modification traces)

Reference

  1. Barrio, R.; Blanco, E.; Parrondo, J.; González, J.; Fernández, J. The effect of impeller cutback on the fluid-dynamic pulsations and load at the blade-passing frequency in a centrifugal pump. Fluids Eng. 2008, 130, 111102.
  2. Parrondo-Gayo, J. L.; Gonzalez-Perez, J.; Fernandez-Francos, J. The effect of the operating point on the pressure fluctuations at the blade passage frequency in the volute of a centrifugal pump. ASME 2002, 124, 784-790.
  3. Shi, L.; Yuan, Y.; Jiao, H.; Tang, F.; Cheng, L.; Yang, F.; Jin, Y.; Zhu, J. Numerical investigation and experiment on pressure pulsation characteristics in a full tubular pump. Energ. 2021, 163, 987-1000.
  4. Zhang, D.S.; Wang, H.Y.; Shi, W.D.; Pan, D.Z.; Shao, P.P. Experimental investigation of pressure fluctuation with multiple flow rates in scaled axial flow pump. Chin. Soc. Agric. Mach. 2014, 45, 139-145.
  5. Li, W.; Ji, L.L.; Shi, W.D.; Zhou, L.; Ping, Y.F. Experiment on pressure fluctuation in mixed-flow pump under different flow rate conditions. Chin. Soc. Agric. Mach. 2016, 47, 70-76.
  6. Zhang, D.S.; Wang, C.C.; Dong, Y.G.; Shi, L.; Jin, Y.X. Tests for inner pressure fluctuation features in an oblique flow pump with high ratio rotating speed. Vib. Shock 2019, 38, 27-34.
  7. Zhu, G. J.; Li, K.; Feng, J.J.; Luo, X.Q. Effects of cavitation on pressure fluctuation of draft tube and runner vibration in a Kaplan turbine. Chin. Soc. Agric. Eng. 2021, 37, 40-49.

Reviewer 3 Report

In this paper, a typical bidirectional axial is taken as the object of study. The hydrodynamic performance of the bidirectional axial flow pump is tested on a high-precision hydro-mechanical test stand, including positive and negative directions. According to the results of the experiment, the hydrodynamic characteristics are comprehensively compared and analyzed, especially the characteristics of the pressure pulsation in the pump, the two-way axial pump in positive and negative operation, as well as the energy characteristics and the law of propagation of pressure pulsations. The work is relevant, but there are comments:
1. The quality of the figures needs to be improved: figures 8, 10, 11, 13, 15, 17.
2. The authors comprehensively analyzed and compared the hydrodynamic characteristics of a bidirectional axial pump in positive and negative operation and the characteristics of pressure pulsations in the pump. The quality of the article only improved if the authors describe in more detail the physical picture and the cause of the appearance of pressure fluctuations.
3. The authors of the article state that: “In recent years, more and more researchers have noted that the pressure pulsation caused by unstable flow in the pump is the most important factor affecting the safety and stability of the pumping station system. Therefore, it is urgent to study the hydrodynamic characteristics of a bidirectional pump, especially the characteristics of internal pressure pulsations.” But they do not give methods and ways to improve the hydrodynamic characteristics, and this only improved the article.
4. It is desirable for the authors to indicate further plans on the topic of the study.

Author Response

Revision explanation

First of all, on behalf of my co-authors, I would like to thank the editors and reviewers for their valuable and constructive comments on our manuscript ' Experimental study on the internal pressure pulsation characteristics of a bidirectional axial flow pump operating in forward and reverse directions '.

We carefully study the reviewers ' comments and modify the manuscript one by one according to each of the comments made by reviewers.

For the comments of reviewers, we modify and supplement the content of the article, and annotate it with red font. Thank the editors and reviewers once again for their valuable comments.

Responses to the comments of reviewer 3

  1. Because we upload PDF files in the submission system before, resulting in the quality of many figures in this paper decreased. Subsequently, we will upload doc format files with original high-quality images, which will significantly improve the quality of images.

  1. We have expanded the analysis content of each chapter according to the suggestions of reviewers, and described and supplemented the causes of physical pictures and pressure fluctuations in more detail, including 4.1,4.2,4.3,4.4 sections. (The contents modified and added in this article have been marked in red fonts to highlight the modification traces)

  1. Thank the reviewers for their valuable suggestions. The main purpose of this paper is to evaluate the safety and stability of the bidirectional axial flow pump system in the bidirectional utilization by using the experimental method. In the following research work, we will do more physical analysis of bidirectional axial flow pump based on CFD method to reveal the failure mechanism of pressure pulsation of bidirectional axial flow pump station under bidirectional operation conditions, and propose methods to eliminate or improve the pressure pulsation of bidirectional axial flow pump.

  1. Based on the recommendations of reviewers, we have added further plans for the topic of this study at the end of the article. (The contents modified and added in this article have been marked in red fonts to highlight the modification traces)

Round 2

Reviewer 2 Report

I have no further comments